# SpikeReveal: Unlocking Temporal Sequences from Real Blurry Inputs with Spike Streams

**Kang Chen**[1,2*]    **Shiyan Chen**[1,2,3*]    **Jiyuan Zhang**[1,2]    **Baoyue Zhang**[1,2]
**Yajing Zheng**[1,2✉]    **Tiejun Huang**[1,2,3]    **Zhaofei Yu**[1,2,3✉]

[1] School of Computer Science, Peking University
[2] State Key Laboratory for Multimedia Information Processing, Peking University
[3] Institute for Artificial Intelligence, Peking University
{mrchenkang,strerichia002p,jyzhang,byzhang}@stu.pku.edu.cn
{yj.zheng,tjhuang,yuzf12}@pku.edu.cn

## Abstract

Reconstructing a sequence of sharp images from the blurry input is crucial for enhancing our insights into the captured scene and poses a significant challenge due to the limited temporal features embedded in the blurry image. Spike cameras, sampling at rates up to 40,000 Hz, have proven effective in capturing motion features and beneficial for solving this ill-posed problem. Nonetheless, existing methods fall into the supervised learning paradigm, which suffers from notable performance degradation when applied to real-world scenarios that diverge from the synthetic training data domain. To address this challenge, we propose the first self-supervised framework for the task of spike-guided motion deblurring. Our approach begins with the formulation of a spike-guided deblurring model that explores the theoretical relationships among spike streams, blurry images, and their corresponding sharp sequences. We subsequently develop a self-supervised cascaded framework to alleviate the issues of spike noise and spatial-resolution mismatching encountered in the deblurring model. With knowledge distillation and reblur loss, we further design a lightweight deblur network to restore high-quality sequences with brightness and texture consistency with the original input. Quantitative and qualitative experiments conducted on our real-world and synthetic datasets with spikes validate the superior generalization of the proposed framework. Our code, data and trained models are available at https://github.com/chenkang455/S-SDM.

## 1 Introduction

Traditional cameras, constrained by their exposure-based imaging mechanism, often produce blurry images when capturing fast-moving objects or during camera movement throughout the exposure process [19, 35]. While these blurry images lose significant details, the ability to recover dynamic motion trajectories from the static blurry input becomes critically important. However, the inherent challenge lies in the limited motion features available within blurry frames, leading to potential ambiguities such as multiple motion trajectories corresponding to the same blurry input. This is exemplified by scenarios where two objects move along the same trajectory but in opposite directions [23, 24, 38], rendering the task of motion deblurring ill-posed. Recent advancements in learning-based approaches [40, 12, 46] seek to address this challenge by establishing direct mappings from

---

[*] Equal contributors.
[✉] Corresponding authors.

38th Conference on Neural Information Processing Systems (NeurIPS 2024).

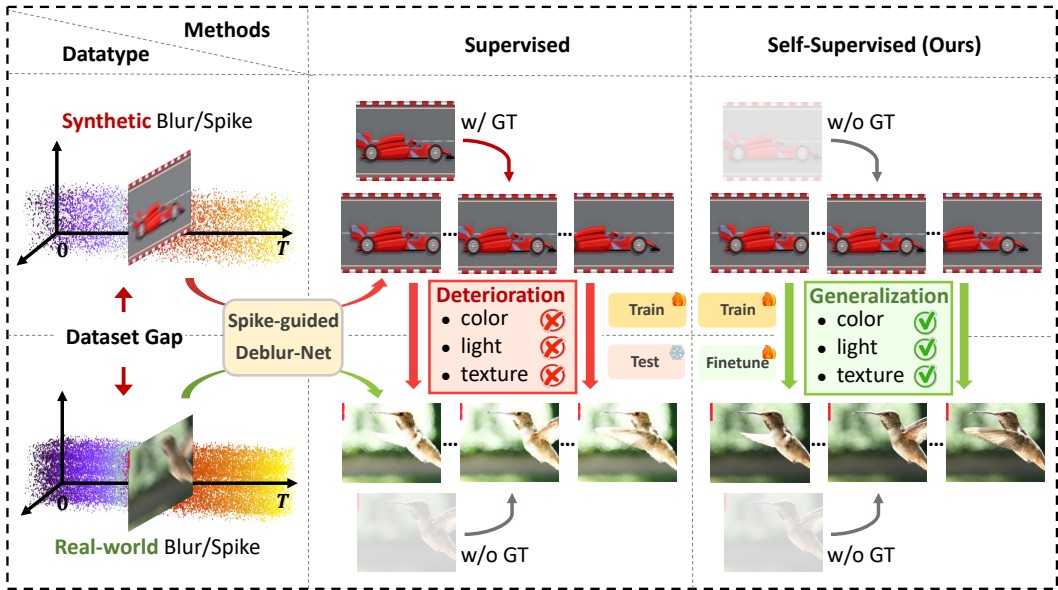

Figure 1: Illustration of the superiority of our self-supervised framework (S-SDM) over supervised methods. Supervised methods, while effective on synthetic datasets, suffer from a significant performance decline when applied to real-world datasets, primarily due to data distribution discrepancies. In contrast, our self-supervised framework, necessitating no Ground Truth (GT) for training, seamlessly bridges this dataset gap through fine-tuning on real-world datasets.

blurry inputs to sharp sequences in a supervised learning manner. Despite these efforts, traditional cameras struggle to capture fine details in high-speed motion due to their exposure constraints, thus limiting the effectiveness of these methods in scenarios not covered by the training datasets.

In recent years, neuromorphic cameras [9, 10], leveraging their ultra-high temporal resolution and high dynamic range, have found widespread use in many fields, including computer vision and robotics. These cameras, including event and spike cameras, are distinguished by their ability to produce high temporal resolution outputs directly tied to changes in light intensity. Specifically, event cameras generate events in areas where light intensity changes [4], while spike cameras capture the absolute brightness of the scene at each pixel, offering a stream of spikes as output [10]. This distinctive feature endows spike cameras with a significant advantage [47, 45, 48, 5, 41, 6] in capturing and recovering sharp texture from scenes with rapid motion.

Recent studies [2, 8] have explored the potential of RGB-Spike fusion, *i.e.*, harnessing the strengths of both traditional and spike cameras to reconstruct sharp sequences from blurry inputs. However, their frameworks are constrained within the supervised learning paradigm, which necessitates extensive datasets comprising pairs of blurry and sharp images, as well as spike sequences. While synthetically acquiring such paired data, as demonstrated in previous studies [8, 4, 24], is feasible, collecting them in real-world scenarios presents the following challenges: (1) high-speed cameras are prohibitively expensive and not readily deployable in many settings; (2) spatial-temporal calibration between spike cameras and high-speed RGB cameras complicates the data collection process. These problems render the fine-tuning of supervised methods on real-world datasets challenging, further leading to their performance deterioration in such environments as shown in Fig. 1. The resulting degradation in image quality, manifesting as color distortion, brightness inconsistency, and inaccurate texture restoration, is mainly caused by the disparity between synthetic and real-world datasets, especially in terms of the density of spike stream, spike generation mechanism, and blurry image generation. Moreover, the effectiveness of supervised methods is inherently limited by the ground truth sequences created through motion analysis interpolation algorithms [22, 11], which inherently differs from the real-world scene and thus affects the model's generalization ability.

To overcome these issues, we propose the first-of-its-kind **S**elf-supervised **S**pike-guided **D**eblurring **M**odel (**S-SDM**), capable of recovering the continuous sharp sequence from a single blurry input with the assistance of low-resolution spike streams. We begin with a theoretical analysis of the

relationship between spike streams, blurry images, and sharp sequences, leading to the development of our Spike-guided Deblurring Model (SDM). We further construct a self-supervised processing pipeline by cascading the denoising network and the super-resolution network to reduce the sensitivity of the SDM to spike noise and its reliance on spatial-resolution matching between the two modalities. To reduce the computational cost and enhance the utilization of spatial-temporal spike information within this pipeline, we further design a Lightweight Deblurring Network (LDN) and train it based on pseudo-labels from the teacher model, *i.e.*, the established self-supervised processing pipeline. Further introducing reblur loss during LDN training, we achieve better restoration performance and faster processing speed than the processing-lengthy and structure-complicated teacher model. To validate the performance of our S-SDM across various scenarios, we build an RGB-Spike binocular system and propose the first spatially-temporally calibrated Real-world Spike Blur (RSB) dataset in this community. Quantitative and qualitative experiments conducted on the real-world and synthetic datasets validate the superiority of our method. In summary, our key contributions are:

- We develop a self-supervised spike-guided image deblurring framework, addressing the performance degradation due to the synthetic-real domain gap in supervised methods.
- We perform an in-depth theoretical analysis of the fusion between the spike stream and blurry image, leading to the development of the SDM.
- We propose a real-world dataset RSB and experiments on GOPRO and RSB datasets validate the superior generalization of our S-SDM.

## 2  Related Work

**Spike Camera.** The spike camera, inspired by the primate retina, stands apart from conventional cameras with its ability to generate synchronous spike streams for each pixel at extremely low latency. This distinct feature provides significant advantages in various applications such as high-speed imaging [47, 45, 48, 5, 41, 6, 33], optical flow estimation [43], object detection [44], 3D reconstruction [30], depth estimation [34], motion deblurring [8, 32], and occlusion removal [31].

**Spike-guided Motion Deblurring.** While the spike camera boasts an ultra-high temporal resolution, its development is currently impeded by the low spatial resolution. Additionally, the single-channel output from the spike camera restricts previous methods from recovering the image color information. To address these issues, a promising approach is establishing an RGB-Spike hybrid imaging system [2]. The binocular system achieves the multi-modality fusion of High-spatial/Low-temporal RGB blurry input and High-temporal/Low-spatial spike stream, thereby also serving as a spike-guided motion deblurring method [8]. However, to the best of our knowledge, existing spike-guided deblurring methods [2, 8] predominantly rely on supervised training on synthetic datasets. This reliance results in significant performance degradation when these methods are evaluated in real-world scenarios due to the domain discrepancies between synthetic and real datasets as illustrated in Fig. 1.

**Event-based Motion Deblurring.** Event camera [21] can asynchronously generate events that record log-intensity changes at the pixel level with minimal latency, which contains a rich set of motion features beneficial for motion deblurring tasks. Numerous supervised methods [16, 4, 3, 16, 23, 24] have been proposed to learn the mapping from the blurry input, events to the sharp outcome. Despite these advancements, a major hurdle remains in obtaining real blurry-sharp image pairs for training. To overcome the domain gap between synthetic and real-world datasets, recent methods [29, 38, 39] explored the mutual constraint between the blurry image and event stream, enabling the training of networks on real-world blur datasets.

## 3  Method

### 3.1  Preliminaries

**Spike Camera Mechanism.** Consider $\mathbf{L}(t)$ to represent the latent sharp frame at time $t$. Each pixel $p$ in the spike camera [10] has an integrator that accumulates the incoming photons at a high frequency. Once the cumulative intensity exceeds a predefined threshold $C$ at time $t_e$, pixel $p$ emits a spike, and the accumulation of photons is reset to zero. This process can be mathematically described as

follows:

$$\int_{t_s}^{t_e} \mathbf{L}(t)dt \geq C, \tag{1}$$

where $t_s$ denotes the firing time of the previous spike. While the spike camera is capable of generating asynchronous spike streams akin to that of event cameras, its effectiveness is constrained by the inherent limitations of its physical circuitry, which necessitates reading spikes at a predetermined sampling rate. We denote the generated spike stream as $\mathcal{S} \in \{0,1\}^{K \times 1 \times H \times W}$, where $H$ and $W$ signify the height and width of the image, and $K$ represents the length of the spike sequence.

**Problem Formulation.** In traditional photography, motion blur occurs when there is relative movement between the camera and the scene during the exposure period. According to the motion blur physical model [9], the blurry image $\mathbf{B}$ can be represented as the average of the latent frame $\mathbf{L}(t)$ over the exposure $\mathcal{T}$, *i.e.*:

$$\mathbf{B} = \frac{1}{T} \int_{t \in \mathcal{T}} \mathbf{L}(t)dt, \tag{2}$$

where $T$ represents the exposure period. Despite the spike camera's superior temporal resolution, its spatial resolution remains comparatively low. This limitation is primarily attributed to the constraints in data transmission bandwidth and the challenges inherent in the manufacturing process. Here, we postulate that the spatial resolution of the spike camera is approximately one-quarter that of a conventional RGB camera.

In this paper, we aim to enhance the High-spatial/Low-temporal resolution blurry input $\mathbf{B} \in \mathbb{R}^{1 \times 3 \times H \times W}$ into a sequence of High-Quality images $\{\mathbf{L}(t_i)\}_{i=1}^{K} \in \mathbb{R}^{K \times 3 \times H \times W}$ with the aiding of High-temporal/Low-spatial resolution spike stream $\mathcal{S}_\mathcal{T} \in \{0,1\}^{K \times 1 \times \frac{H}{4} \times \frac{W}{4}}$, which can be mathematically formulated as:

$$\{\mathbf{L}(t_i)\}_{i=1}^{K} = \text{Deblur}(t_i; \mathbf{B}, \mathcal{S}_\mathcal{T}). \tag{3}$$

In Eq. (3), $\text{Deblur}(\cdot)$ represents the Spike-guided Deblur-Net as shown in Fig. 1, $i$ refers to the $i$-th frame in the spike stream $\mathcal{S}_\mathcal{T}$, and $t_i$ is the timestamp associated with this frame.

## 3.2 Theoretical Analysis

The spike camera, with its photodetector tailored to capture the single-channel light intensity, faces difficulties in obtaining the color information that the multi-channel RGB camera can effortlessly capture. Therefore, we modify the color intensity $\mathbf{L}(t)$ in Eq. (1) to the grayscale value $\mathbf{L}_g(t)$ for further analysis:

$$\int_{t_s}^{t_e} \mathbf{L}_g(t)dt \geq C. \tag{4}$$

In this formulation, $\mathbf{L}_g(t) = w_r \cdot \mathbf{L}_r(t) + w_{gre} \cdot \mathbf{L}_{gre}(t) + w_b \cdot \mathbf{L}_b(t)$, where $\mathbf{L}_c(t), w_c$ denote the intensity and weight of channel $c \in \{r, gre, b\}$ respectively.

Given the blurry input $\mathbf{B}$ and its corresponding spike stream $\mathcal{S}_\mathcal{T}$, we incorporate Eq. (4) into the motion blur model presented in Eq. (2). This integration formulates a link between the two modalities as outlined below:

$$\mathbf{B}_g = \frac{C \cdot N_\mathcal{T}}{T}, \tag{5}$$

where $\mathbf{B}_g$ denotes the grayscale version of the blurry input, and $N_\mathcal{T}$ denotes the total number of spikes accumulated over the exposure period. The accumulation $N_\mathcal{T}$ is calculated as $N_\mathcal{T} = \sum_{i=1}^{K} \mathcal{S}[i]$, with $\mathcal{S}[i]$ indicating the $i$-th frame of the spike stream.

Within the exposure $\mathcal{T}$, we consider a shorter spike sequence centered around the $t$ moment $\mathcal{S}_{\mathcal{T}'} \in \{0,1\}^{K' \times 1 \times H \times W}$, satisfying $K' \ll K, t \in \mathcal{T}$, and $\mathcal{T}' \subset \mathcal{T}$. Similar to Eq. (5), we can derive the relationship between the short-exposure gray image $\mathbf{E}_g(t, \mathcal{T}')$ and the short spike stream $\mathcal{S}_{\mathcal{T}'}$ as follows:

$$\mathbf{E}_g(t, \mathcal{T}') = \frac{1}{T'} \int_{s \in \mathcal{T}'} \mathbf{L}_g(s)ds = \frac{C \cdot N_{\mathcal{T}'}}{T'}, \tag{6}$$

where $T' \ll T$ represents the short exposure period.

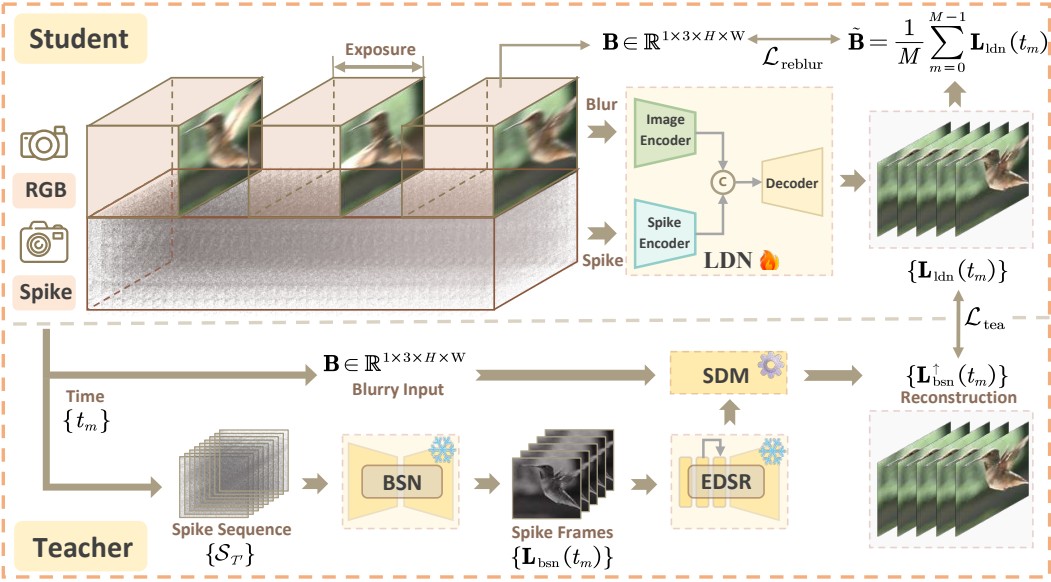

Figure 2: The schematic diagram of our proposed distillation self-supervised framework. The "⚙" indicates that certain computations are executed in a non-network manner.

Given the observation that the color information of adjacent pixels often exhibits similarity under the premise of minor motion amplitude, we postulate that the colors of the blurry input $\mathbf{B}$ and the short-exposure image $\mathbf{E}(t, \mathcal{T}')$ are identical. This assumption implies that the intensity proportion among RGB channels in the blurry input $\alpha_c^{\mathbf{B}}$ and the short-exposure image $\alpha_c^{\mathbf{E}}(t, \mathcal{T}')$ is approximately equivalent, satisfying $\alpha_c^{\mathbf{B}} = \mathbf{B}_g / \mathbf{B}_c$ and $\alpha_c^{\mathbf{E}}(t, \mathcal{T}') = \mathbf{E}_g(t, \mathcal{T}') / \mathbf{E}_c(t, \mathcal{T}')$. More details can be found in the supplementary materials.

Upon establishing it, we move forward to build a mathematical relation between the blurry image and the spike stream. By substituting the gray channel $g$ with color channel $c$ and dividing Eq. (5) by Eq. (6), we efficiently eliminate the unknown threshold $C$ and weights $\alpha_c^{\mathbf{B}} / \alpha_c^{\mathbf{E}}(t, \mathcal{T}')$, leading to the following equation:

$$\mathbf{E}_c(t, \mathcal{T}') = \mathbf{B}_c \cdot \frac{N_{\mathcal{T}'}}{N_{\mathcal{T}}} \cdot \frac{T}{T'}. \tag{7}$$

By applying Eq. (7) to RGB three channels, we explicitly establish the relationship between the color blurry input $\mathbf{B}$, the color short-exposure image $\mathbf{E}(t, \mathcal{T}')$ and the spike stream $\mathcal{S}_{\mathcal{T}}$ as shown in Fig. 18. Since the exposure period $\mathcal{T}'$ is relatively short, it is reasonable to assume that the scene remains static. In this context, we interpret the short-exposure image $\mathbf{E}(t, \mathcal{T}')$ as the latent sharp frame $\mathbf{L}(t)$, allowing us to modify Eq. (7) as follows:

$$\mathbf{L}(t) = \mathbf{B} \cdot \frac{N_{\mathcal{T}'}}{N_{\mathcal{T}}} \cdot \frac{T}{T'}. \tag{8}$$

To this end, we have conducted a comprehensive theoretical analysis of the spike-guided motion deblurring task, which is neglected in prior learning-based motion deblur methodologies [2, 8]. For further discussion readability, we refer to Eq. (8) as the **Spike-guided Deblurring Model (SDM)**, which is analogous to the baseline motion deblur model EDI [18] in event camera.

### 3.3 Self-supervised Spike-guided Deblurring Model

#### 3.3.1 Processing Pipeline

While SDM theoretically allows for the fusion of the blurry image and the spike stream, its practical deployment faces the following obstacles:

- The deblurred image $\mathbf{E}(t, \mathcal{T}')$ suffers from noise-related degradation due to the lack of adequate spike information during the short exposure $\mathcal{T}'$.

- The spatial resolution of the spike camera is approximately one-quarter of the RGB camera, rendering the SDM implementation impractical.

To overcome these limitations, we further cascade the self-supervised denoising network to eliminate the spike noise in $N_{\mathcal{T}'}$ and super-resolution network to match spatial resolutions of the blurry image and spike stream, with the processing pipeline illustrated in the bottom of Fig. 2.

**Denoising Network.** We leverage the Blind Spot Network (BSN) [13, 1, 27, 5, 14, 7] to predict the clean spike accumulation $N_{\mathcal{T}'}$ from the input short-exposure spike stream $\mathcal{S}_{\mathcal{T}'}$. The core idea of BSN is to design the blind-spot strategy that compels the convolutional layer to estimate the clean value of each pixel solely based on its surrounding pixels.

Under the premise that the spike stochastic thermal noise is independent identically distributed [42], the BSN is trained to deduce sharp spike frames from the input, with the loss function formulated as:

$$\mathcal{L}_{\text{BSN}} = ||\text{BSN}(\mathcal{S}_{\mathcal{T}'}; \Theta_1) - N_{\mathcal{T}'})||_2^2, \tag{9}$$

where the denoised spike frame $\text{BSN}(\mathcal{S}_{\mathcal{T}'}; \Theta_1)$ is denoted by $\mathbf{L}_{\text{bsn}}(t)$ for further analysis.

**Super-Resolution Network.** In this task, we observe that the blurry input $\mathbf{B}_g$ and the long-exposure spike frame $N_{\mathcal{T}}$ exhibit the same texture features as shown in Eq. (5). This observation motivates us to train the Super-Resolution (SR) network based on pairs of the blurry images and the long-exposure spike frames.

We leverage the well-explored Enhanced Deep Super-Resolution network (EDSR) [15] as the backbone of our SR network, with the loss function formulated as follows:

$$\mathcal{L}_{\text{EDSR}} = ||\text{EDSR}(N_{\mathcal{T}}; \Theta_2) - \mathbf{B}_g||_2^2. \tag{10}$$

With the training of the SR network completed, we freeze its parameters and apply it to the denoised spike frame $\mathbf{L}_{\text{bsn}}(t)$, yielding the resolution-enhanced spike frame $\mathbf{L}_{\text{bsn}}^{\uparrow}(t)$.

### 3.3.2 Knowledge Distillation Framework

While the aforementioned processing pipeline achieves the multi-modality fusion of the blurry input and the spike stream, several aspects still need refinement:

- The framework is lengthy and computationally demanding, which hinders its suitability for real-time system deployment.
- The blind-spot strategy of the BSN limits the full utilization of the spatial information inherent in the spike stream.
- The representation of the short-exposure image does not fully reflect the advantages of the high temporal resolution inherent in the spike stream.

To improve them, we further build a knowledge distillation framework building upon the existing processing pipeline. This pipeline serves as the teacher model, providing the reconstructed sequence as pseudo-labels for the training of the student model LDN, as illustrated in Fig. 2.

**Lightweight Deblur Network.** LDN adheres to a similar input and output pattern as previous research [8], *i.e.*, taking the blurry input $\mathbf{B}$ and the short spike stream $\mathcal{S}_{\mathcal{T}'}$ centered around moment $t$ as inputs, with the output being the reconstructed sharp image $\mathbf{L}_{\text{ldn}}(t)$, mathematically formulated as follows:

$$\mathbf{L}_{\text{ldn}}(t) = \text{LDN}(\mathbf{B}, \mathcal{S}_{\mathcal{T}'}; \Theta_3), \tag{11}$$

Full details regarding the LDN structure are available in the supplementary materials.

To avoid the scenario where the LDN exactly replicates the mapping of the teacher model, we design the teacher loss $\mathcal{L}_{\text{tea}}$ based on the LPIPS [36] loss and further introduce the blur reconstruction loss. The reblur loss $\mathcal{L}_{\text{reblur}}$ measures the difference between the blurry input $\mathbf{B}$ and the re-synthesized blurry image $\widetilde{\mathbf{B}}$, satisfying:

$$\widetilde{\mathbf{B}} = \frac{1}{M} \sum_{m=1}^{M} \mathbf{L}_{\text{ldn}}(t_m), \tag{12}$$

where $\mathbf{L}_{\text{ldn}}(t_m)$ represents the $m$-th recovered image within the exposure period $\mathcal{T}$ and $M$ is the total number of reconstructed images. Finally, we sum up two loss functions with the weighting parameter

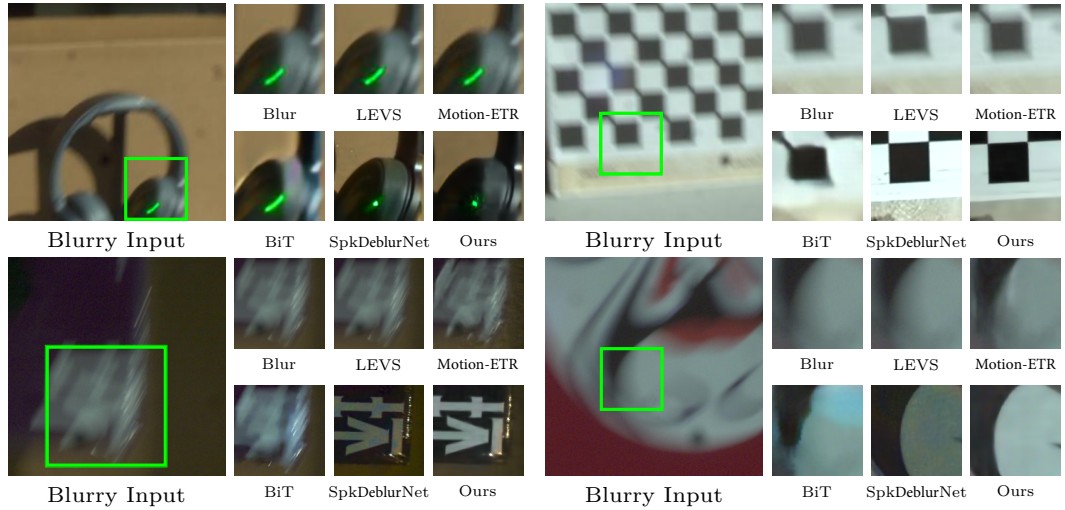

Figure 3: Qualitative comparison for the single frame restoration on the RSB dataset.

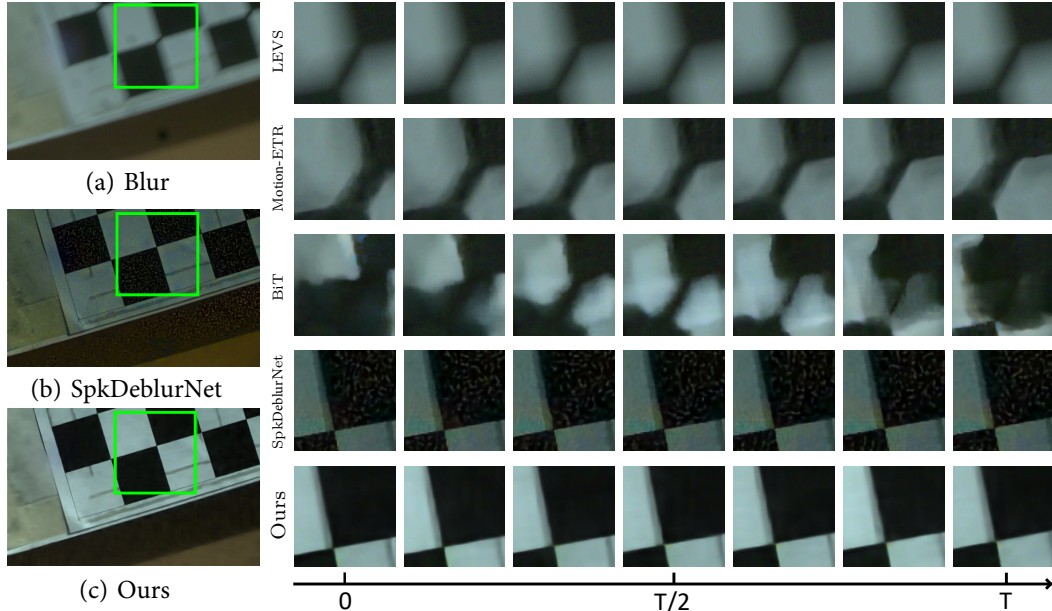

Figure 4: Qualitative comparison for the sequence reconstruction on the RSB dataset.

$\lambda$, and the final loss function is formulated as follows:

$$\mathcal{L} = \mathcal{L}_{\text{tea}} + \lambda \cdot \mathcal{L}_{\text{reblur}} \tag{13}$$

$$= \sum_{m=1}^{M} \mathcal{L}_{\text{LPIPS}}(\mathbf{L}_{\text{bsn}}^{\uparrow}(t_m), \mathbf{L}_{\text{ldn}}(t_m)) + \lambda \cdot \mathcal{L}_{\text{MSE}}(\widetilde{\mathbf{B}}, \mathbf{B}). \tag{14}$$

## 4 Experiments

### 4.1 Dataset

**Synthetic Data.** For quantitative analysis of our spike-guided motion deblurring task, we construct the synthetic dataset based on the widely employed GOPRO [17] dataset. We initially utilize the interpolation algorithm XVFI [22] to augment the video frame by interpolating additional 7 frames between each pair of consecutive sharp images. To generate the spike stream that mimics reality closely, we downsample the interpolated video to the resolution of $320 \times 180$ and simulate the spike

Table 1: Quantitative comparison of the sequence reconstruction task on the GOPRO dataset.

| Methods | Spike | $V_{th}$=1 | | $V_{th}$=2 | | $V_{th}$=4 | | Params |
| --- | --- | --- | --- | --- | --- | --- | --- | --- |
| | | PSNR | SSIM | PSNR | SSIM | PSNR | SSIM | |
| LEVS[12] | × | 21.155 | 0.601 | 21.155 | 0.601 | 21.155 | 0.601 | 4.97M |
| Motion-ETR[40] | × | 21.955 | 0.610 | 21.955 | 0.610 | 21.955 | 0.610 | 6.55M |
| BiT[46] | × | 23.644 | 0.698 | 23.644 | 0.698 | 23.644 | 0.698 | 11.3M |
| TRMD[4]+DASR[26] | ✓ | 27.323 | 0.784 | 21.198 | 0.601 | 18.567 | 0.523 | 19.3M |
| RED[29]+DASR[26] | ✓ | 24.456 | 0.741 | 23.178 | 0.674 | 21.942 | 0.608 | 9.76M |
| REFID[25]+DASR[26] | ✓ | 28.124 | 0.819 | 15.288 | 0.339 | 13.623 | 0.274 | 15.9M |
| SpkDeblurNet[8] | ✓ | **28.307** | **0.834** | 14.406 | 0.299 | 11.621 | 0.202 | 13.4M |
| S-SDM (Ours) | ✓ | 26.893 | 0.757 | **26.367** | **0.740** | **25.433** | **0.699** | **0.23M** |

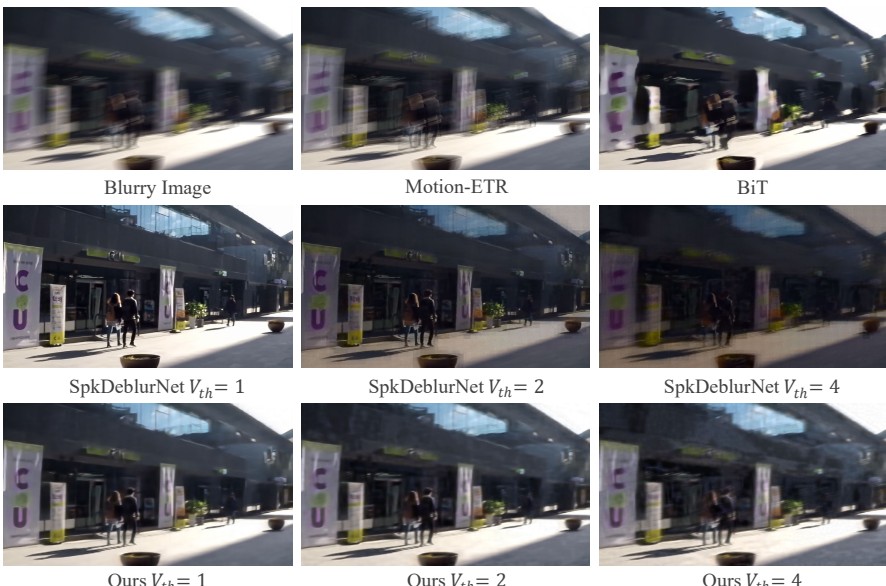

Figure 5: Visual comparison of our S-SDM against other methods on the GOPRO dataset.

stream based on the spike simulator [42]. To replicate real-world motion blur, we synthesize each blurry input by averaging 97 frames from interpolated video sequences.

**Real-world Data.** We construct an RGB-Spike binocular system and propose the first **R**eal-world **S**pike-guided **B**lur dataset (**RSB**) in this community. This system consists of a fixed-exposure RGB camera (Basler acA1920-150uc) and a spike camera [10], enabling us to capture the blurry image and the corresponding spike stream simultaneously. Further details about our RSB and the spatial-temporal calibration for our binocular system are provided in the supplementary materials.

## 4.2 Experimental Results

We conduct both quantitative and qualitative comparisons of our S-SDM against state-of-the-art (SOTA) motion deblurring methods, including frame-based Motion-ETR [40], LEVS [12], video-based BiT [46], event-based TRMD [4], REFID [25], RED [29] and spike-based SpkDeblurNet [8] on the GOPRO and RSB datasets. For event-based methods, we replace the event stream with the spike stream and adopt the same input representation in these methods [29, 25, 4]. We further cascade the image super-resolution technique DASR [26] as in [39] for the deblurred sequence to overcome the modality resolution inconsistency which is not considered in these methods. We reconstruct 7 images from one blurry input for sequence restoration evaluation [39] as listed in Tab. 1.

Table 2: Performance comparison between the SAN in GEM [39] and our designed LDN.

| Methods | PSNR ↑ | SSIM ↑ | Params (M) ↓ | Flops (G) ↓ |
|---|---|---|---|---|
| SAN [39] | 27.283 | 0.773 | 2.36 | 107.84 |
| LDN (Ours) | **27.928** | **0.786** | **0.234** | **33.60** |

**Results on GOPRO.** To simulate the spike density domain gap as depicted in Fig. 1, we train all supervised spike-based deblurring methods on the GOPRO dataset under spike threshold [42] $V_{th} = 1$ and evaluate them on datasets with spike thresholds $V_{th} = 1, 2, 4$. Quantitative comparison results are listed in Tab. 1 and the visual comparison is demonstrated in Fig. 5. Given that the principal contribution of our S-SDM is self-supervised learning, it is foreseeable that our method might be slightly inferior to the supervised methods SpkdDeblurNet on the dataset with $V_{th} = 1$. While these supervised methods deteriorate on datasets with $V_{th} \neq 1$, our method achieves great generalization benefiting from the self-supervision design. Specifically, SpkDeblurNet tends to produce darker and blurrier reconstructions on images with high thresholds as shown in Fig. 5. Besides, our method achieves better restoration performance than the self-supervised method RED due to our consideration of the spatial-resolution mismatch between two modalities and the designed teacher loss, which imposes a stronger constraint than the optical loss in RED.

**Results on RSB.** We further present visualizations of single frame and sequence reconstruction comparisons on the real-world RSB dataset as depicted in Fig. 3 and 4. Both frame-based and video-based approaches fail to replicate fine textures and detailed elements present in the blurry input. While SpkDeblurNet is capable of recovering structural details and the motion trajectory, it is deteriorated by significant noise and color distortion under conditions with lower spike density than those simulated in the GOPRO dataset. Similarly, in scenarios of higher spike stream densities, the restoration of SpkDeblurNet tends to exhibit over-exposure, resulting in brightness inconsistency compared to the blurry input. This over-exposure affects the dynamic range of the reconstructed image, ultimately compromising the overall perceptual quality and uniformity of the restored sequence. Our method addresses these challenges by finetuning on the RSB dataset, ensuring that the restored sequence aligns with the real-captured blurry input and spike stream. More comparative experiments and analyses are accessible in the supplementary material.

## 4.3 Ablation Study

We perform ablation experiments on the GOPRO and RSB datasets to evaluate the performance of each module within S-SDM, the validity of our designed network architecture, as well as the overall effectiveness of our distillation learning framework. In this subsection, we evaluate the performance based on the single middle frame for simplicity.

**Modules Cascading.** Building upon the SDM, we sequentially cascade the BSN, the SR, and the LDN to evaluate their respective effectiveness, with quantitative results on GOPRO presented in Tab. 3. We employ bilinear interpolation as a substitute for the SR network in experiments I-1 and I-2 to align the spatial resolution of two modalities.

Qualitative ablation experiments are illustrated in Fig. 6. These comparisons reveal that while the SDM effectively removes motion blur, it struggles with significant noise and detail loss due to the spike noise and the low resolution of the spike stream. While the BSN mitigates noise and the SR network improves spatial resolution explicitly, the LDN trained via distillation learning further refines these enhancements, enabling the recognition of intricate textural features in the images, such as the license plate and the door number shown in Fig. 6.

**Network Architecture.** While our designed LDN mirrors the Scale-aware Network (SAN) proposed in the GEM [39], we replace the SAN with the LDN to compare the performance difference between the two architectures as depicted in Tab. 2. Despite the simple design of our LRN, which consists only of convolutional layers, ResBlocks, and basic modules such as CBAM [28], it outperforms SAN in both PSNR and SSIM while requiring fewer parameters and less computation, demonstrating that LDN is both sufficient and efficient for this task. This conclusion is consistent with our previous discussion, *i.e.*, the restoration performance of the self-supervised learning framework primarily depends on the quality of pseudo-labels rather than the network architecture.

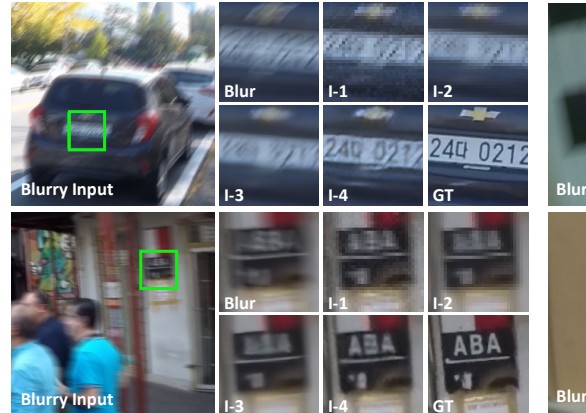
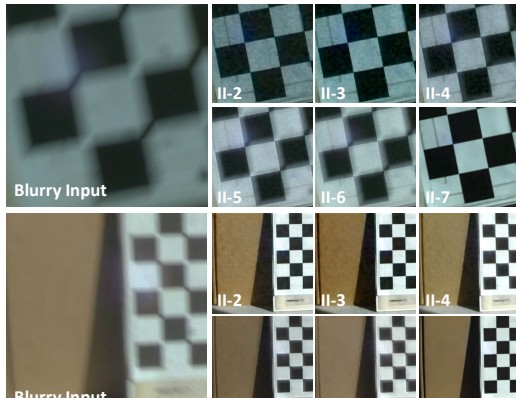

Figure 6: Modules cascading comparisons on GO-PRO. Experiments ID can be viewed on Tab. 3.

Figure 7: Distillation learning comparisons on RSB. Experiments ID can be viewed on Tab. 4

Table 3: Modules cascading ablation on GOPRO.

| ID | BSN | SR | LDN | PSNR↑ | SSIM↑ |
|---|---|---|---|---|---|
| I-1 | × | | | 23.012 | 0.486 |
| I-2 | ✓ | | | 24.634 | 0.661 |
| I-3 | ✓ | ✓ | | 26.144 | 0.708 |
| I-4 | ✓ | ✓ | ✓ | **27.928** | **0.786** |

Table 4: Distillation learning ablation on GOPRO.

| ID | $\mathcal{L}_{\text{tea}}$ | $\mathcal{L}_{\text{reblur}}$ | $\lambda$ | PSNR↑ | SSIM↑ |
|---|---|---|---|---|---|
| II-1 | × | ✓ | 10 | 23.102 | 0.441 |
| II-2 | ✓ | × | / | 26.563 | 0.723 |
| II-3 | ✓ | ✓ | 10 | 27.345 | 0.762 |
| II-4 | ✓ | ✓ | 50 | 27.742 | 0.778 |
| II-5 | ✓ | ✓ | 100 | **27.928** | **0.786** |
| II-6 | ✓ | ✓ | 200 | 27.620 | 0.783 |

**Distillation Learning.** We focus on analyzing the contribution of the teacher loss $\mathcal{L}_{\text{tea}}$ and the reblur loss $\mathcal{L}_{\text{reblur}}$ within the distillation framework, with quantitative results listed in Tab. 4. Without the teacher loss $\mathcal{L}_{\text{tea}}$, the LDN tends toward learning identity mapping from the blurry input. While under the guidance of the teacher model, the reblur loss $\mathcal{L}_{\text{reblur}}$ not only enforces motion consistency in the reconstructed sequence but also enriches the LDN with high-resolution details from the non-blurry regions of the input, thus improving the performance on GOPRO as listed in Tab. 4.

We further apply the LDN trained on GOPRO to the real-world dataset RSB, with qualitative visualization illustrated in Fig. 7. As observed in the figure, the absence of the reblur loss $\mathcal{L}_{\text{reblur}}$ leads to significant noise in the recovered image, which predominantly arises from the disparity in spike density and generation mechanism between the simulated and real-captured spike stream. This discrepancy causes the LDN to overestimate the spike number, resulting in black holes in regions with lower spike density than simulated, which reflects the drawback inherent in the supervised learning strategies as discussed in Sec. 1. Increasing the weight of the reblur loss $\mathcal{L}_{\text{reblur}}$ allows the LDN to incorporate more information from the blurry input, thereby mitigating this issue. However, this adjustment also leads to the presence of blurry edges. We follow the parameters set in Experiment II-6 and retrain the LDN on the RSB dataset (referred to as Experiment II-7). The retrained LDN effectively recovers the sharp edge of the calibration board and suppresses the spike noise in the background, validating the feasibility of our self-supervised framework in real-world scenarios.

## 5 Conclusion

In this paper, we introduce a novel self-supervised spike-guided motion deblurring framework S-SDM, which reconstructs sequences of sharp images from real-world blurry inputs with the spike stream. Additionally, we construct an RGB-Spike binocular system and propose the first spatially-temporally calibrated real-world dataset RSB in this community. Quantitative and qualitative experiments validate the superior generalization capabilities of our proposed S-SDM.

**Limitation.** The limitation of our S-SDM lies in its dependence on strict spatial-temporal calibration. Misalignment will lead to color shifts and quality degradation in the deblurred sequence.

## Acknowledgments

We sincerely appreciate Yuyan Chen (HUST) for her valuable suggestions and for polishing the figures. This work was supported by the National Natural Science Foundation of China (62176003, 62088102, 62306015), the China Postdoctoral Science Foundation (2023T160015), the Young Elite Scientists Sponsorship Program by CAST (2023QNRC001), and the Beijing Nova Program (20230484362).

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

Figure 8: Video comparison of our method against the BiT and SpkDeblurNet on the RSB dataset under different luminance conditions, with the static visualization displayed in Fig. 13. It is recommended to view the pdf using Acrobat PDF reader and the gif demonstration with a higher frame rate is available in the supplementary zip file.

## A Appendix

This supplementary material provides a comprehensive elaboration on the methodologies and experiments in this paper. It is organized into four distinct sections: Theory Analysis in appendix A.1, Network Settings in appendix A.2, RSB Dataset in appendix A.3, Experimental Details in appendix A.4 and Additional Figure in appendix A.5.

### A.1 Theory Analysis

We define $k_1^{\mathbf{B}}$ and $k_2^{\mathbf{B}}$ as the ratios of the red channel to the green and blue channels in the blurry input $\mathbf{B}$ respectively, *i.e.*,

$$k_1^{\mathbf{B}} = \mathbf{B}_r/\mathbf{B}_{gre}, \tag{15}$$

$$k_2^{\mathbf{B}} = \mathbf{B}_r/\mathbf{B}_b. \tag{16}$$

Since the gray image is the weighted sum of RGB channels, the ratio of the gray to the red image $\alpha_r^{\mathbf{B}}$ is formulated as:

$$\alpha_r^{\mathbf{B}} = \mathbf{B}_g/\mathbf{B}_r \tag{17}$$

$$= (w_r \cdot \mathbf{B}_r + w_{gre} \cdot \mathbf{B}_{gre} + w_b \cdot \mathbf{B}_b)/\mathbf{B}_r \tag{18}$$

$$= w_r + w_{gre}/k_1^{\mathbf{B}} + w_b/k_2^{\mathbf{B}}, \tag{19}$$

where $w_c$ denote the weight of channel $c \in \{r, gre, b\}$ respectively.

Similarly, we define $k_1^{\mathbf{E}}(t, \mathcal{T}')$ and $k_2^{\mathbf{E}}(t, \mathcal{T}')$ as the fractions of the red channel relative to the green and blue channels in the short-exposure image $\mathbf{E}(t, \mathcal{T}')$, resulting in:

$$\alpha_r^{\mathbf{E}}(t, \mathcal{T}') = w_r + w_{gre}/k_1^{\mathbf{E}}(t, \mathcal{T}') + w_b/k_2^{\mathbf{E}}(t, \mathcal{T}'). \tag{20}$$

Given the observation that the color information of adjacent pixels often exhibits similarity under the premise of minor motion amplitude, we postulate that the colors of the blurry input $\mathbf{B}$ and the short-exposure image $\mathbf{E}(t, \mathcal{T}')$ are identical. This assumption implies that the intensity proportion among RGB channels in two images is approximately equivalent, *i.e.*, $k_1^{\mathbf{B}} \approx k_1^{\mathbf{E}}(t, \mathcal{T}')$ and $k_2^{\mathbf{B}} \approx k_2^{\mathbf{E}}(t, \mathcal{T}')$. Drawing from Eq. (19) and Eq. (20), We further deduce the following relation:

$$\alpha_r^{\mathbf{B}} \approx \alpha_r^{\mathbf{E}}(t, \mathcal{T}'), \tag{21}$$

which can be readily generalized across channels, leading to $\alpha_c^{\mathbf{B}} \approx \alpha_c^{\mathbf{E}}(t, \mathcal{T}')$.

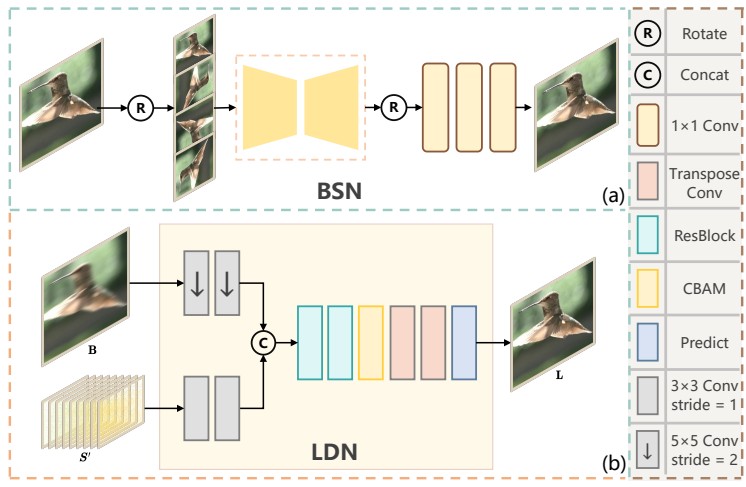

Figure 9: Network diagrams of the BSN (a) and our designed LDN (b).

## A.2 Network Settings

### A.2.1 Blind Spot Network

We construct our BSN based on the blind-spot strategy outlined in [13]. We first rotate the input image four times, then concatenate them into the U-Net [20] structure like denoising network. To prevent direct mapping from the input pixel to the output pixel, a single-pixel offset strategy is employed in the convolutional kernel to separate its receptive field from the central pixel. Finally, the denoising output is obtained by merging the results of four branches via a $1 \times 1$ convolution, as shown in Fig. 9(a).

### A.2.2 Lightweight Deblur Network

The network structure is depicted in Fig. 9(b), where the encoder for blurry images consists of two layers of down-sampling convolutions to align the spatial resolution of two modalities. In contrast to the intricate cross-attention mechanism outlined in [4, 8], we implement the fusion of two modalities by the simple operation of $\text{Concat}(\cdot)$, with the cascaded CBAM [28] and residual blocks for further feature fusion.

## A.3 RSB Dataset

We detail the construction of our RGB-Spike hybrid system as illustrated in Fig. 10. The system comprises a Spike Camera-001T-Gen2 with a resolution of $400 \times 250$ pixels, paired with a Basler acA1920-150uc RGB Camera, offering a higher resolution of $1920 \times 1200$ pixels.

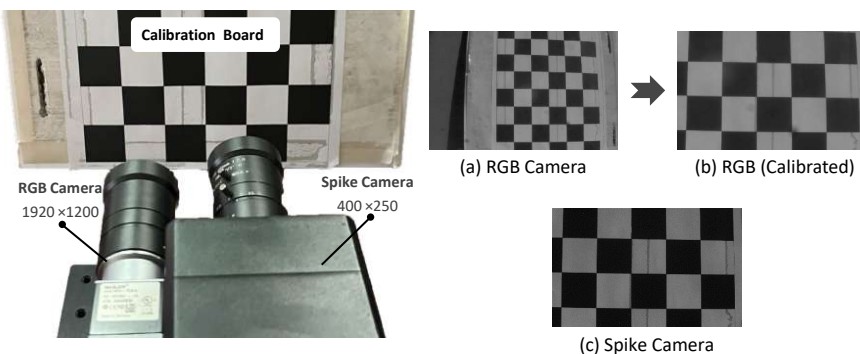

Figure 10: RGB-Spike camera system.

Figure 11: Calibration result.

Table 5: Quantitative comparison of the single frame task on the RSB dataset, where non-reference metric LIQE ranging from 1 to 5 is employed. LIQE is a positive metric denoted as ↑ where higher scores reflect better performance.

| Methods | Board-L | Board-M | Board-H | Face | Earphone | Average ↑ |
|---|---|---|---|---|---|---|
| LEVS [12] | 1.0010 | 1.0342 | 1.0400 | 1.0029 | 1.1382 | 1.0433 |
| Motion-ETR [40] | 1.0153 | 1.0064 | 1.0279 | 1.0023 | 1.1724 | 1.0449 |
| BiT [46] | 1.0002 | 1.0069 | 1.0473 | 1.0138 | 1.6155 | 1.1367 |
| SpkDeblurNet [8] | 1.8493 | 1.3232 | 2.2698 | 1.2210 | 2.6606 | 1.8648 |
| S-SDM (Ours) | **2.0955** | **1.3872** | **2.4208** | **1.3814** | **2.7087** | **2.1987** |

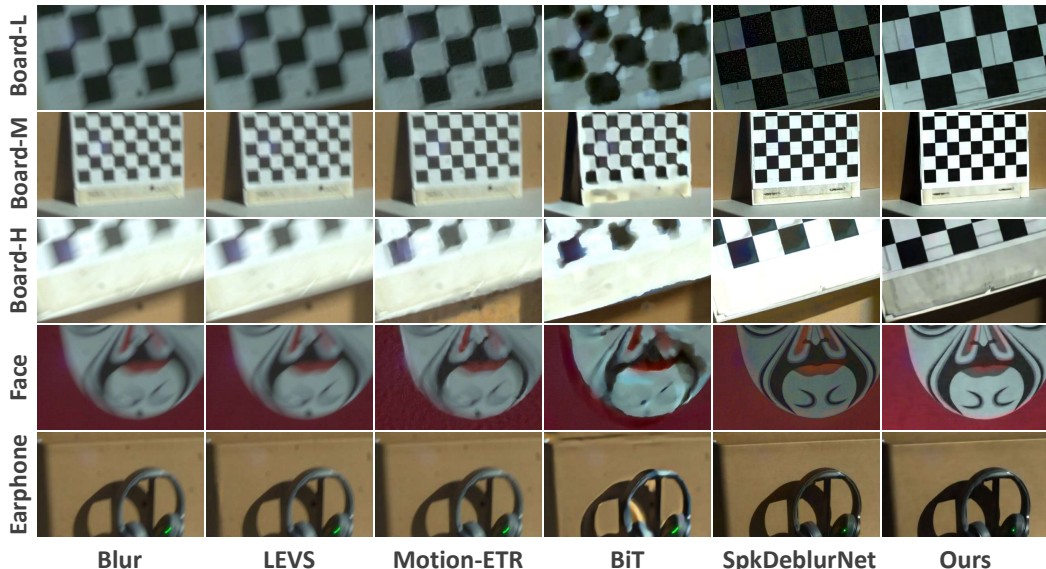

Figure 12: Qualitative comparisons for single-frame restoration on the RSB dataset are illustrated, where "Board-L", "Board-M", and "Board-H" represent the board captured under low, middle and high lighting conditions.

To achieve the spatial calibration of two cameras, we perform simultaneous captures of the calibration board using the RGB-Spike system as shown in Fig. 10. The RGB images are cropped to a resolution of 1600 × 1000, aligning them to be fourfold the resolution of the spike camera. We convert the cropped RGB image to grayscale and take the TFP [47] image reconstruction result as the reference from the spike camera. We utilize the MATLAB calibration toolbox to implement the calibration process, with the results detailed in Fig. 11.

Our RSB dataset contains 10 video sequences under different conditions, captured under varied conditions including scene brightness levels (*e.g.*, Low, Middle, and High light) and motion patterns (*e.g.*, camera shake and object motion), which introduce different types of motion blur. Besides, the RSB dataset comprises a large amount of blur-spike pairs with each blurry input corresponding to 400 spike frames.

## A.4 Experimental Details

### A.4.1 Comparison

To assess the performance of our method on the GOPRO dataset, we utilize the Peak Signal-to-Noise Ratio (PSNR) and the Structural Similarity Index (SSIM) as the quantitative metrics, which are commonly used in motion deblurring tasks. In real-world datasets, where obtaining ground truth sharp sequence is challenging, we opt for the non-reference image quality assessment method Language-Image Quality Evaluator (LIQE) [37] as a reference. By assessing visual quality through

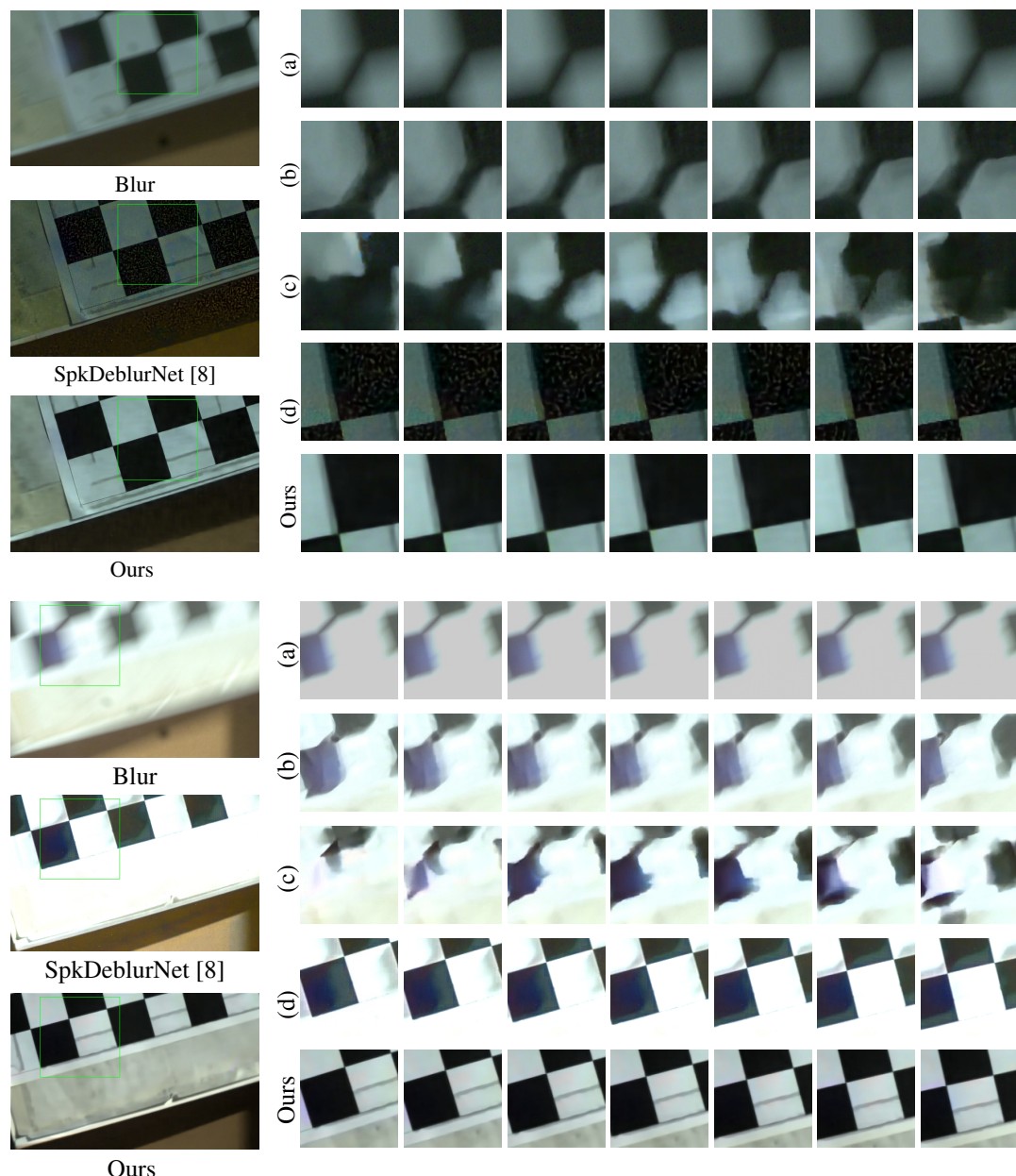

Figure 13: Qualitative comparison for the sequence reconstruction on the RSB dataset. (a),(b),(c),(d) denote results of "LEVS", "Moiton-ETR" , "BiT" and "SpkDeblurNet" respectively. The upper panel depicts a real-world scene with a lower spike density than the simulation, whereas the lower image exhibits a higher spike density.

the computation of joint probabilities from visual-textual embeddings, LIQE adeptly identifies the clarity and blurriness of images independently of the ground truth, making it ideally suited for our task.

**Comparison on the RSB dataset.** Qualitative and quantitative experiments of the single frame restoration task on the RSB dataset are shown in Fig. 12 and Tab. 5 respectively. The visual results coupled with the LIQE metrics demonstrate that our method outperforms other methods in handling the real-world RSB dataset. While the supervised SpkDeblurNet encounters substantial noise and overexposure challenges in both low-light and high-light environments, our approach demonstrates superior restoration performance, which is attributed to the designed self-supervised framework.

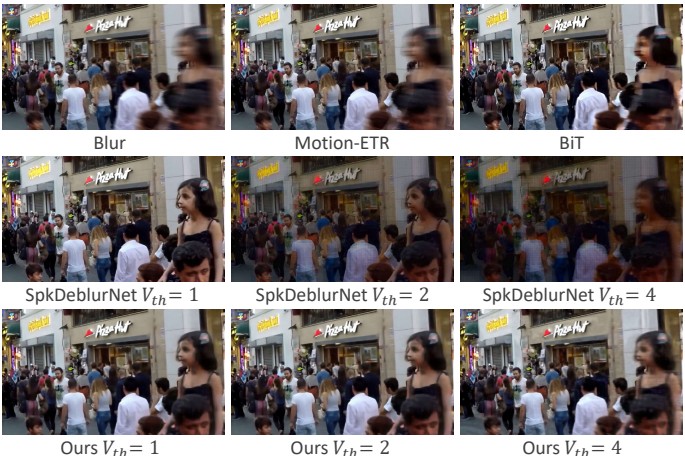

Figure 14: Comparison of our S-SDM against other methods on GOPRO with $V_{th} = 1, 2, 4$.

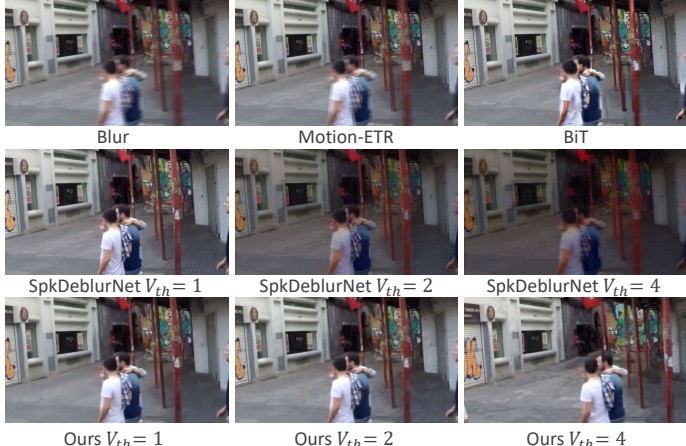

Figure 15: Comparison of our S-SDM against other methods on GOPRO with $V_{th} = 1, 2, 4$.

Moreover, as depicted in Fig. 13 and Fig. 8, S-SDM exhibits outstanding performance in precisely retrieving luminance information and texture details while ensuring the motion consistency of reconstructed sequences. In contrast, the video reconstruction of the BiT suffers from poor image quality and shows inadequate sequence continuity in the restored sequence. Besides, SpkDeblurNet encounters issues with color distortion, brightness inconsistency, and inaccurate texture restoration owing to the domain gap between synthetic and real-world datasets. These observations further highlight the superior performance of our S-SDM in real-world scenarios.

**Comparison on the GOPRO dataset.** We provide additional visual comparison of our method against other SOTA methods on the GOPRO dataset as shown in Fig. 14 and 15.

### A.4.2 Implementation

To augment the dataset and accelerate the training process, we randomly crop $512 \times 512$ image from each blurry frame, along with the $128 \times 128$ spike stream. We use PyTorch to build and train our S-SDM using an NVIDIA GeForce GTX 4090 GPU and AMD EPYC 7742 64-Core Processor. The training of our LDN consumes about 4 hours on the GOPRO. During the testing phase, we feed the entire image and the spike stream into the network to assess performance.

We complete the training of BSN on the GOPRO dataset, employing an initial learning rate of $3e^{-4}$ and spanning 1000 epochs. The training uses the Adam optimizer with a cosine scheduler and sets the batch size to 8 for each epoch. Adopting the same settings as BSN, EDSR is trained on the blur-spike

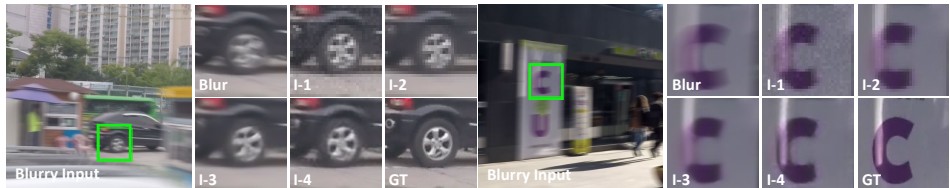

Figure 16: Ablation study for evaluating modules of S-SDM on the GOPRO dataset. Experiments corresponding to the ID can be viewed through Tab. 3.

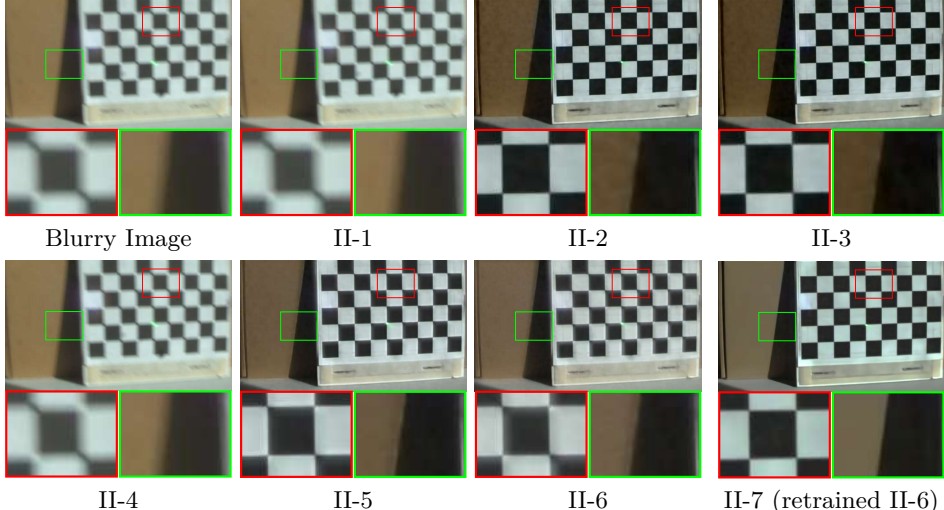

Figure 17: Ablation study for evaluating the effectiveness of our distillation learning framework on the RSB dataset. The details corresponding to the experiment ID can be viewed through the Tab. 4.

paired data for 70 epochs. Subsequently, LDN undergoes the training of 100 epochs with the learning rate adjusted to $1e^{-3}$.

### A.4.3 Ablation study

We provide additional ablation visualizations to demonstrate the effectiveness of our designed modules and the distillation framework as shown in Fig. 16 and 17.

### A.5 Additional Figure

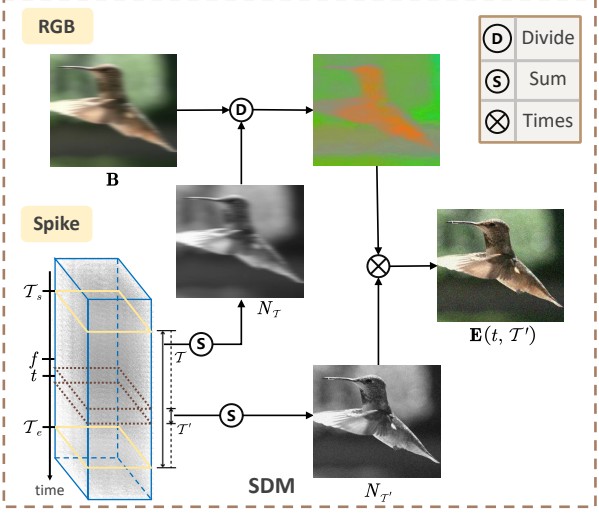

Figure 18: The schematic diagram of our designed SDM.

