# OpenReview forum: "SpikeReveal: Unlocking Temporal Sequences from Real Blurry Inputs with Spike Streams"
_NeurIPS.cc/2024/Conference — NeurIPS 2024 spotlight_

### Official Review · Reviewer_1hhy · 2024-07-07

**Soundness:** 3
**Presentation:** 3
**Contribution:** 3
**Rating:** 6
**Confidence:** 4

**Summary:**

The SpikeReveal paper uses a blurry RGB image and leverages pulse data captured by a spike camera of the corresponding scene to guide the image deblurring task. In terms of network design, the authors cascade a blind-spot network for denoising, a super-resolution network, and a deblurring network, employing a self-supervised approach to accomplish the deblurring task.

**Strengths:**

The SpikeReveal paper uses a blurry RGB image and leverages pulse data captured by a spike camera of the corresponding scene to guide the image deblurring task. In terms of network design, the authors cascade a blind-spot network for denoising, a super-resolution network, and a deblurring network, employing a self-supervised approach to accomplish the deblurring task.

**Weaknesses:**

The paper has several areas that could be improved to better achieve its stated goals.  Addressing these issues would improve the robustness and validity of the comparative analysis.

**Questions:**

[Definition of t_s]
There is an issue with the definition of t_s in line 109. When the first spike occurs, there is no previous spike to reference. Therefore, this definition needs to be revised.
[Missing Ablation Study]
Table 2: The module cascading ablation on GOPRO should include with BSN without SR with LDN and without BSN with SR with LDN. This would allow us to determine which component, BSN, SR, or LDN, contributes the most to the final performance.
[Comparison with SpkDeblurNet]
In Table 1 and Figure 4, has SpkDeblurNet been retrained using the same simulation data as the authors' method? Why do the restoration results of SpkDeblurNet in Figure 4 appear noticeably darker? Additionally, why does SpkDeblurNet perform better than the authors' method at Vth=1V_{th}=1?

**Limitations:**

The authors could consider collecting paired datasets to better address the limitations of simulating pulse camera data. While the authors mention using real data, it serves only as input to the network without providing a genuine ground truth for comparison. Addressing this by capturing paired real-world data could significantly enhance the validity and applicability of their findings.

---

> ### Author Rebuttal · Authors · 2024-08-03
>
> We are grateful for your detailed feedback and suggestions, which have helped us identify key areas where our manuscript can be improved.
>
> ***1.  [Definition of t_s] There is an issue with the definition of t_s in line 109. When the first spike occurs, there is no previous spike to reference. Therefore, this definition needs to be revised.***
>
> Thank you for pointing out this issue. We will address the problem you mentioned in the final version.
>
>  ***2. [Missing Ablation Study] Table 2: The module cascading ablation on GOPRO should include with BSN without SR with LDN and without BSN with SR with LDN. This would allow us to determine which component, BSN, SR, or LDN, contributes the most to the final performance.***
>
> We have supplemented the ablation study based on your request, including the configurations for BSN + LDN and SR + LDN. The overall results of the ablation study are as follows:
>
> **Table.R5: Ablation study on the proposed different modules.**
> | ID           | BSN | SR  | LDN | PSNR   | SSIM  |
> |:--------------------:|:-------------------:|:-------------------:|:-----------:|:--------------:|:-------------:|
> | I-1          | ✗   | ✗   | ✗   | 23.012 | 0.486 |
> | I-2          | ✓   | ✗   | ✗   | 24.634 | 0.661 |
> | I-3          | ✓   | ✓   | ✗   | 26.144 | 0.708 |
> | I-4     | ✗   | ✓   | ✓   | 26.172 | 0.633 |
> | I-5   | ✓   | ✗   | ✓   | 26.662 | 0.745 |
> | I-6          | ✓   | ✓   | ✓   | **27.928** | **0.786** |
>
> The table compares the performance of various configurations involving BSN, SR and LDN in terms of PSNR and SSIM. The baseline configuration (I-1), with no methods applied, shows the lowest image quality metrics. Adding BSN alone (I-2) significantly enhances both PSNR and SSIM, demonstrating effective noise reduction. When SR is added (I-3), further improvement is observed, reflecting enhanced detail recovery. The BSN + LDN setup (I-5) provides better results than SR + LDN (I-4), confirming BSN’s key role in enhancing quality over the SR network, which will amplify the image noise without the pre-processing of the BSN network thus degrading the image quality and reconstruction performance.
> Finally, the full combination of BSN, SR, and LDN (I-6) delivers the highest PSNR and SSIM scores, showcasing the synergistic effect of integrating all three methods for optimal image deblurring performance.
>
> ***3. [Comparison with SpkDeblurNet] In Table 1 and Figure 4, has SpkDeblurNet been retrained using the same simulation data as the authors' method? Why does SpkDeblurNet perform better than the authors' method at Vth=1V_{th}=1?***
>
> Fig. 4 presents the comparison of different methods in the sequence reconstruction task for real-world scenarios. In real scenes, the absence of ground truth sharp frames means that the SpkDeblurNet method cannot be retrained, leading to noise-related image degradation. In contrast, the S-SDM method effectively overcomes the domain gap and restores sharp texture details, benefiting from its self-supervised mechanism. Both S-SDM and SpkDeblurNet are trained under the same simulation dataset while S-SDM is further fine-tuned in real-world scenarios.
>
> In Tab. 1, to mimic the domain gap between synthetic and real datasets and quantitatively represent this gap using PSNR and SSIM metrics, we designed our experiments such that all methods were trained with a scenario where $V_{th} = 1$ and evaluated across scenarios with different threshold values. The reason SpkDeblurNet performs better under the $V_{th} = 1$ condition is that it is constrained by strong supervision signal, which makes the network to learn the mapping from the blurry input and the spike stream to the sharp image. Under this specific setting, the strength of our self-supervised framework, S-SDM, to effectively overcome the domain gap is not fully realized.
>
>  ***4. Why do the restoration results of SpkDeblurNet in Figure 4 appear noticeably darker?***
>
> Please refer to the response in **To all reviewers**.
>
>  ***5. The authors could consider collecting paired datasets to better address the limitations of simulating pulse camera data. While the authors mention using real data, it serves only as input to the network without providing a genuine ground truth for comparison. Addressing this by capturing paired real-world data could significantly enhance the validity and applicability of their findings.***
>
> Simultaneously capturing spike streams, blurry image inputs, and clear images requires precise spatiotemporal calibration of three different cameras. This process is time-consuming and labor-intensive, with the potential for calibration inaccuracies. Thank you for your feedback. We will consider further improving our camera system in future research to develop a dataset that includes RGB inputs, spike streams, and ground truth clear images.

---

### Official Review · Reviewer_RXBh · 2024-07-07

**Soundness:** 2
**Presentation:** 3
**Contribution:** 2
**Rating:** 5
**Confidence:** 4

**Summary:**

The work focuses on improving image sharpness from blurry inputs using spike cameras with high-motion capture rates. Addressing limitations of supervised learning in real-world scenarios, the authors introduce a self-supervised framework for spike-guided motion deblurring. Validation through extensive experiments on both real-world and synthetic datasets confirms its superior performance and generalization ability.

**Strengths:**

This paper introduces a pioneering approach to spike-guided motion deblurring, addressing the challenge of recovering sharp images from blurry inputs captured by spike cameras. It demonstrates high-quality research with rigorous theoretical foundations and thorough experimental validations on synthetic and real-world datasets.

**Weaknesses:**

1，I'm curious about how the order of applying BSN (Blur to Sharp Network) and EDSR (Enhanced Deep Super-Resolution) influences model performance. Specifically, I wonder if this sequence could introduce additional artifacts during image restoration.

**Questions:**

see the weakness

**Limitations:**

yes

---

> ### Author Rebuttal · Authors · 2024-08-03
>
> Thank you for dedicating your time to provide constructive criticism and recommendations for our article.
>
> ***I'm curious about how the order of applying BSN (Blur to Sharp Network) and EDSR (Enhanced Deep Super-Resolution) influences model performance. Specifically, I wonder if this sequence could introduce additional artifacts during image restoration.***
>
> Before answering this question, we need to clarify why we need the BSN and EDSR networks. As stated in our paper, the SDM model has two main issues:
>
> 1. Short-exposure spike frames contain **significant noise** due to limited information contained during this period.
>
> 2. Two modalities have **inconsistent resolutions**, making it challenging to apply the SDM model directly to real-world scenarios. To address these issues, the self-supervised denoising network BSN and the super-resolution network EDSR are employed. The EDSR network is retrained based on the texture similarity between long-exposure spike frame and blurry input image.
>
> In the following, we first explain the difference of BSN-EDSR and EDSR-BSN two orders from the following perspectives:
>
> ***A. Theoretical Analysis***
>
> **EDSR-BSN:**  The advantage of this order is that it can recover as much information as possible from the low-resolution details, making it suitable for spike reconstructions that are rich in detail and have less noise. However, a drawback is that the super-resolution process may amplify the original noise in the image, making it difficult for subsequent denoising to completely remove the noise.
>
> **BSN-EDSR:**  The advantage of this order is that the denoised image provides a sharper baseline for super-resolution, helping the algorithm more accurately reconstruct image details and avoid amplifying noise. The drawback is that if the denoising is too aggressive, it may remove important details from the image, making it challenging for super-resolution to recover these details, especially in the image where details are relatively limited.
>
> ***B. Quantitative Experiments***
>
> In the GOPRO dataset, both scenarios mentioned above are present.  Through evaluating the performance of both orders in the entire dataset, we find that the denoising-first, super-resolution-second approach (BSN-EDSR) yields better results, so we adpot this processing baseline.
>
> **Table.R4: Ablation study on the order of the BSN and EDSR networks.**
> | Method  | PSNR   | SSIM  |
> |:---------:|:--------:|:-------:|
> | EDSR-BSN  | 25.914 | 0.692 |
> | BSN-EDSR  | **26.144** | **0.708** |
>
>
> ***C. Visual Comparison***
>
> To understand this intuitively, we conduct a visual comparison on the GOPRO dataset, as shown in Fig. R3. In the example (street scene) above Fig. R3, it can be seen that the EDSR-BSN method effectively recovers some detailed features of the floor compared to the BSN-EDSR method. The EDSR-BSN method cannot accurately recover these details because the BSN network first removes them, leaving little information for the EDSR network to super-resolve.
>
> In the example below, regarding some background information, the EDSR-BSN method produces noticeable noise and image artifacts. (The magnified image regions use contrast enhancement to highlight detailed information.) This occurs because applying the EDSR to the reconstructed spike frame first enlarges the noise, resulting in a noisy final image. The subsequent BSN network cannot mitigate this due to the limited neighboring information available during the blind convolution process.

---

### Official Review · Reviewer_VbQb · 2024-07-11

**Soundness:** 4
**Presentation:** 4
**Contribution:** 4
**Rating:** 7
**Confidence:** 5

**Summary:**

This work proposes a spike-guided self-supervised image deblurring algorithm that combines the high spatial resolution of RGB cameras with the high temporal resolution of spike cameras to obtain sharp RGB images in real-world scenarios. The self-supervised network addresses performance degradation issues found in existing supervised spike deblurring algorithms. The spike-guided module considers noise and spatial resolution alignment between the two cameras. Both subjective and quantitative experimental results show that the proposed algorithm achieves excellent generalization performance in both real and simulated scenes, surpassing previous spike-guided deblurring algorithms.

**Strengths:**

1. The proposed self-supervised spike-guided RGB deblurring algorithm effectively addresses the synthetic-real domain gap performance degradation of previous algorithms.
2. The authors analyze the relationships between spike data, blurry RGB, and sharp RGB images, providing a theoretical foundation for the Spike-Guided Deblurring Model (SDM).
3. The paper's clear writing and figures make it easy to follow, with comprehensive experiments demonstrating the RGB-Spike binocular system's spatiotemporal alignment and generalization in real scenarios.

**Weaknesses:**

1. While many existing image-based deblurring algorithms perform adequately, the introduction of spike cameras presents challenges such as aligning the two modalities. The authors have not sufficiently compared or explained the advantages of the spike-guided approach.
2. The robustness of the model to noise lacks systematic analysis and discussion. Is this primarily influenced by the BSN Loss described in Equation 9?

**Questions:**

This method primarily targets optical flow motion estimation. For more complex motion In addition to addressing the performance degradation issues of some supervised models in real-world scenarios, what other advantages does the proposed unsupervised model offer?

**Limitations:**

The authors have stated the limitations.

---

> ### Author Rebuttal · Authors · 2024-08-03
>
> Thank you for your thoughtful review and for pointing out potential issues and improvements in our paper.
>
> ***1. While many existing image-based deblurring algorithms perform adequately, the introduction of spike cameras presents challenges such as aligning the two modalities. The authors have not sufficiently compared or explained the advantages of the spike-guided approach.***
>
> We aim to address this question from the following perspectives:
>
> **A. Theoretical Analysis**. Image-based methods for addressing motion blur often struggle to capture motion information accurately in real-world scenes because traditional cameras cannot precisely record motion details during the exposure. This limitation can lead to incorrect motion trajectories in real-world scenarios. For example, when using a conventional camera to capture a square object moving from left to right, the resulting blurry image does not inherently indicate whether the object is moving from left to right or right to left without prior information. Additionally, reconstructing the texture details of an object at any moment during the exposure from a single blurred image is difficult because it is an ill-posed problem with limited motion representation. However, spike streams, which contain rich motion and texture information, can effectively alleviate issues related to uncertain motion trajectories and missing texture details.
>
> **B. Experimental Results.** On the synthetic dataset GOPRO, image-based methods like BiT have achieved promising results. This is because the motion patterns in the GOPRO dataset are relatively consistent, and the methods for synthesizing motion blur are also uniform. However, as seen in Fig. 3, 4, 11, and Tab. 4, in real-world scenarios where the motion patterns differ significantly from those in the GOPRO dataset, the BiT algorithm encounters severe image degradation and inaccuracies in motion trajectory recovery, which is evident in both quantitative performance results and qualitative visual outcomes.
>
> **C. Ablation Study.** We conduct a simple ablation study using SpkDeblurNet on the GOPRO dataset in SpkDeblurNet to verify the contribution of spike streams in assisting deblurring. The comparison results are shown in the Tab. R3. It can be seen that the incorporation of the spike stream can improve about 5 db in PSNR for the single frame image deblurring task, which will behave better in the sequence reconstruction task since image-based methods suffer from severe motion ambiguity.
>
> **Table.R3: Ablation study on the effectiveness of the spike stream.**
> | Method        | PSNR  | SSIM  |
> |:---------------:|:-------:|:-------:|
> | Image         | 32.45 | 0.895 |
> | Image + Spike | **37.42** | **0.968** |
>
>
>  ***2. The robustness of the model to noise lacks systematic analysis and discussion. Is this primarily influenced by the BSN Loss described in Equation 9?***
>
> Please refer to the response in **To all reviewers**.
>
>  ***3. For more complex motion In addition to addressing the performance degradation issues of some supervised models in real-world scenarios, what other advantages does the proposed unsupervised model offer?***
>
> The core contribution of this paper is the introduction of a spike-based deblurring physical model, SDM, and a self-supervised spike-based deblurring model, S-SDM. The S-SDM model primarily addresses the noise and resolution mismatch issues present in the SDM model. In addition to overcoming the domain gap problem, the proposed unsupervised model has the following advantages:
>
> **A. Interpretability.** Our SDM model theoretically constructs the relationship between blurred images, spike streams, and sharp images. Compared to previous fully end-to-end spike deblurring networks like SpkDeblurNet, our method has a stronger theoretical foundation.
>
> **B. Implementability.** The SDM is a model-based motion deblurring method, which can be directly deployed in real-time systems and is relatively easy to implement.
>
> **C. Deployability.** Our S-SDM features a core motion deblurring network, LDN, with a very small parameter size (0.23M). Compared to the parameters of other supervised learning networks like SpkDeblurNet, the LDN network's parameters are only 2% of SpkDeblurNet's, making it more suitable for deployment in real-world motion deblurring scenarios.

---

> > ### Comment · Reviewer_VbQb · 2024-08-12
> >
> > Thanks to the authors for their response and the insights on the model's noise robustness, as well as its advantages over other image-based deblurring algorithms. The additional explanations provided by the authors regarding the S-SDM method from the perspectives of Interpretability, Implementability, and Deployability were also very clear. I have no further questions.
> > I raised my score to 7.

---

> > > ### Author Response · Authors · 2024-08-12
> > > **Thanks for your valuable time and increasing the score.**
> > >
> > > Thank you sincerely for your insightful comments, valuable suggestions, kind appreciation of our work and increasing the score. Thanks a lot for your valuable time! Your time and input mean a lot to us.

---

### Official Review · Reviewer_G1PY · 2024-07-13

**Soundness:** 3
**Presentation:** 4
**Contribution:** 3
**Rating:** 6
**Confidence:** 5

**Summary:**

This paper combines the RGB camera and the spike camera for image deblur. The key contributions consist a self-supervised learning framework for deblur and a real-world dataset RSB.

**Strengths:**

This paper presents a novel self-supervised framework for image deblur with spike camera. The network design is interesting. The paper is well-written.

**Weaknesses:**

1 The originality is marginal. The whole framework is similar to [36]. Please clarify more on the difference between this paper and [36].

2. The proposed LDN is also similar to the DCN in [36]. It would be better to see in the ablation whether LDN is better than DCN.

3. It can be observed that smaller V_th leads to better performance. What if V_th == 0.5?

4. The size and diversity of the RSB dataset is limited. It seems the RSB dataset only contains indoor scenes. Outdoor scenes with various objects are desired.

**Questions:**

See the weaknesses


------------------------------------
The response addressed my concerns. I raised my score to 6.

**Limitations:**

See the weaknesses

---

> ### Author Rebuttal · Authors · 2024-08-03
>
> We sincerely appreciate the time and effort you have taken to review our manuscript.
>
> ***1. The originality is marginal. The whole framework is similar to [36]. Please clarify more on the difference between this paper and [36].***
>
> Paper \[36\] is an ICCV23 publication on a self-supervised evnet-based motion deblurring algorithm named GEM. Besides both being self-supervised algorithms, the content and framework designs of S-SDM and GEM are completely different. We explain the differences between the two in the following aspects:
>
> **A. Camera Principles**: GEM is designed for event cameras while our S-SDM is applied on spike cameras. These two neuromorphic cameras leverage different sampling techniques—differential sampling for event cameras and integral sampling for spike cameras. This difference results in substantial variation in the framework of the two self-supervised motion deblurring methods.
>
> **B. Number of Blurry Frames**: GEM relies on two blurry frames to provide mutual information for constraint when designing the self-supervised loss function, as shown in Fig. R2. However, S-SDM only requires a single blurry frame to complete the self-supervised task, making it more suitable for deployment in online real-time motion deblurring systems compared to GEM.
>
> **C. Self-Supervised Pipeline**: The core of GEM’s self-supervised algorithm is to explore the relationship between two different blurry images, Blur1 and Blur2, with the aid of the event stream to achieve self-supervised deblurring. In contrast, the core of S-SDM’s self-supervised approach is to enhance the quality of pseudo-labels using a teacher network consisting of the SDM, BSN, and EDSR, while exploring the relationship between the latent sharp image and the input blurry image to achieve self-supervision, as shown in Fig. R2.
>
> **D. Loss Functions**: GEM’s core loss function is the Blur2Blur loss, which trains the network to learn Blur2 from Blur1 with the aid of the event stream. S-SDM’s core loss function is the blur consistency reblur loss, which ensures that the deblurred images obtained at different times during the exposure time, when recombined, are consistent with the original blurry image, as shown in Fig. R2.
>
> **E. Teacher Model**: GEM uses the SAN network as the teacher model for generalizing it to scenes at different spatiotemporal scales, while S-SDM’s teacher model consists of SDM, BSN, and EDSR, aiming to provide high-quality pseudo-labels for the LDN network.
>
> **F. Theoretical Analysis**: GEM analyzes the relationship between varying degrees of blurriness and event streams. In contrast, S-SDM studies single-frame blurry images, spike streams, and sharp images. Due to the different sampling principles of spike cameras and event cameras, the theoretical analyses of GEM and S-SDM are entirely different.
>
>  ***2. The proposed LDN is also similar to the DCN in [36]. It would be better to see in the ablation whether LDN is better than DCN.***
>
> We should first clarify that the network framework in the GEM paper is the Scale-aware Network (SAN) rather than DCN. DCN is merely a feature extraction module used to enhance feature extraction in the image domain.
>
> Regarding the issue mentioned, we want to clarify that the core contribution of this paper lies in designing a self-supervised processing pipeline for deblurring with spike cameras. **The network design is not the core contribution. As stated in the paper, we use the encoder-fusion-decoder multimodal fusion framework in designing the network in line with previous studies.** This framework has been widely used in many previous studies. In the SAN network, the core design is not the encoder-fusion-decoder framework but rather its unique MSFF block.
>
> **Why didn’t we explore further in network architecture?** The reason is that for the self-supervised learning framework designed in this paper, improving network performance depends mainly on how to enhance the quality of pseudo-labels. In the supervised learning framework spk2deblurnet, since the supervision signal is direct and explicit, exploring the network architecture to better fit the relationships between blurry images, spike inputs, and sharp images can effectively improve performance. However, for a self-supervised pipeline, a lightweight and simple network is sufficient. Therefore, the network in this paper consists only of convolutional layers, ResBlocks, and simple modules like CBAM.
>
> Finally, to prove our statement, we compare our designed LDN network with the SAN network designed in GEM in terms of PSNR, SSIM, Params, and Flops on the single-frame motion deblurring task:
>
> **Table.R2: Comparison between the SAN network in GEM and our designed LDN.**
> | Methods | PSNR  | SSIM  | Params (M) | Flops (G)   |
> |:---------:|:-------:|:-------:|:------------:|:-------------:|
> | SAN [36]    | 27.283| 0.773 | 2.36   | 107.84|
> | LDN (Ours)    | **27.928**| **0.786**| **0.234**    | **33.60** |
>
> As shown in Tab. R2, for the self-supervised deblurring task, our LDN network achieves better performance in terms of PSNR and SSIM while maintaining smaller model parameters and computational requirements. This demonstrates that for the self-supervised spike deblurring task presented in this paper, our designed simple network LDN is sufficient for this task while being parameters and computation lightweight.
>
>  ***3. It can be observed that smaller V_th leads to better performance. What if V_th == 0.5?***
>
> Please refer to the response in **To all reviewers**.
>
>  ***4. The size and diversity of the RSB dataset is limited. It seems the RSB dataset only contains indoor scenes. Outdoor scenes  are desired.***
>
> Our RSB dataset comprises 10 video sequences, including 9 indoor scenes and 1 outdoor scene as shown in Fig. R4. Due to the time-consuming nature of time and space calibration across two modalities, we plan to collect more scenes and increase the diversity, which will be included in the submission version.

---

> > ### Comment · Reviewer_G1PY · 2024-08-12
> >
> > Thanks for the response.
> > I raised my score to 6.

---

> > > ### Author Response · Authors · 2024-08-12
> > > **We greatly appreciate your thoughtful feedback.**
> > >
> > > We're especially grateful for the increased score and the time you've dedicated to reviewing our paper. Your insights and support are truly meaningful to us. Thank you!

---

### Author Rebuttal · Authors · 2024-08-03

We thank all reviewers for their constructive comments and positive feedback. We are pleased that our paper has been recognized as "well-written" [G1PY,VbQb] with a "pioneering approach" [RXBh] and our network design found "interesting" [G1PY]. Our proposed S-SDM is acknowledged for effectively addressing the synthetic-real domain gap [VbQb], providing a strong theoretical foundation [VbQb], and demonstrating excellent experimental results [VbQb, RXBh]. We are also grateful for the recognition of our method to synergistically enhance the deblurring results in a self-supervised manner [1hhy]. The reviewers' insights are invaluable, and we will integrate all suggestions to refine our work further.

In this general response, we would like to address the crucial concerns regarding '***How $V_{th}$ influences the experiments.***'

> **[Reviewer G1PY]**: It can be observed that smaller V_th leads to better performance. What if V_th == 0.5?

The threshold $V_{th}$ has a multifaceted impact in this task. When the spike threshold is high, the spike firing rate of the spike camera decreases, but the dark current and other noise in the spike camera simulation remain unchanged. This results in a lower signal-to-noise ratio  of the input information, leading to poorer deblurring performance. Conversely, when the spike threshold is too low, the spike camera's synchronous sampling mechanism causes issues. A lower threshold leads to multiple spikes being fired between intervals, but only one is read, resulting in less spike stream information during the readout time.

Considering this problem from an extreme perspective: if the spike firing threshold is infinitely high, there will be no spike readout, only dark current noise. If the spike firing threshold is infinitely low, every sampling cycle will produce frames filled with ones, making it impossible to extract meaningful information.

We further evaluated the deblurring performance of the S-SDM model in a sequence recovery task under different spike threshold values of 0.25 and 0.5. The experimental results are as follows:

**Table.R1: Comparison of our method under different $V_{th}$.**
| $V_{th}$| 0.25   | 0.5    |    1     | 2      | 4      |
|:--------:|:--------:|:--------:|:--------:|:--------:|:--------:|
| PSNR   | 24.657 | 25.886 | 26.893 | 26.367 | 25.433 |
| SSIM   | 0.633  | 0.724  | 0.757  | 0.740  | 0.699  |

From the above table, it can be seen that $V_{th}= 1$ achieves the best performance, which aligns well with our previous analysis. Additionally, we provide visualization comparison of the SDM model at a threshold of $V_{th} = 0.5$ in Fig. R1. It can be observed that even though the short exposure spike frame suffers from texture damage due to truncation errors in most regions, the SDM model effectively retains the correct texture information by utilizing the blurry image $B$. SDM model, while removing motion blur from the blurry input, significantly mitigating truncation errors caused by the low threshold in the spike frames, which further demonstrates the robustness of our method.

> **[Reviewer VbQb]**: The robustness of the model to noise lacks systematic analysis and discussion. Is this primarily influenced by the BSN Loss described in Equation 9?

The proposed method effectively alleviates degradation issues such as noise, primarily due to the design of its self-supervised framework. This design grants it strong generalization capabilities in both real-world and synthetic datasets, as demonstrated in Fig. 1 of this paper. We agree with your view that the robustness of our method against noise is largely attributed to the BSN loss function. Next, we will explain the role of the BSN loss from the following perspectives:

**A. Working Mechanism.** The robustness of the BSN model against noise is primarily due to its use of a blind-spot convolution strategy. This approach employs self-supervision to eliminate noise in the short-exposure spike imaging frames. As a result, the BSN model can effectively remove noise of any type through retraining, offering strong generalization capabilities.

**B. Visual Comparison:** From the ablation studies in Figs. 5 and 17, we can see that I-2 effectively eliminates the substantial background noise present in I-1, such as the noise around license plates and letter signs. This demonstrates the effectiveness of the BSN self-supervised loss. This loss function is also capable of effectively removing dark current noise present in real-world scenarios, thus ensuring generalization to real scenes.

**C. Quantitative Analysis:** According to our latest ablation experiment results in Tab. R5, the BSN loss function effectively removes background noise both when used alone and when combined with SR and LDN. It improves PSNR by 1.6 dB and 1.8 dB, respectively, in these configurations.

> **[Reviewer 1hhy]** Why do the restoration results of SpkDeblurNet in Figure 4 appear noticeably darker?

The reason SpkDeblurNet appears significantly darker in Fig. 4 is due to its nature as a supervised learning method. It has only learned to extract deblurring texture details from the spike streams with fixed firing rates in the synthetic GOPRO dataset ($V_{th} = 1$ under this condition). When applied to real-world scenarios, if the spike firing rate in the real scene differs from that of the synthetic dataset, SpkDeblurNet misestimates the spike density. This results in darker outputs when the spike density is lower than in the synthetic dataset, and overexposure in scenarios where the spike density is higher, as seen in Fig. 12.

---

### Author Response · Authors · 2024-08-11
**Look Forward to Feedbacks**

Dear Reviewers,

Thank you sincerely for your review. We would greatly appreciate it if you could inform us of any remaining questions or concerns that you may have, so that we can address them promptly prior to the deadline. Alternatively, if you feel that your initial concerns are addressed, we would appreciate updating your evaluation to reflect that. Thank you!

---

### Decision · Program_Chairs · 2024-09-25

**Decision:**

Accept (spotlight)

**Comment:**

The paper was rated by the reviewers with an Accept, 2 Weak Accept, and a Borderline Accept. The reviewers raised a number of concerns, but they were all addressed by the authors and the reviewers raised their initial ratings.
There is general consensus that the paper introduces a novel approach to deblur images by exploiting the spike camera.
The AC has read all the reviews, the rebuttal, and the discussions and agrees with the reviewers that the paper should be accepted.